# Technical note: Quantified organic aerosol subsaturated hygroscopicity by a simple optical scatter monitor system through field measurements

Jie Zhang[1], Tianyu Zhu[1,2], Alexandra Catena[1], Yaowei Li[3], Margaret J. Schwab[1], Pengfei Liu[4], Akua A. Asa-Awuku[5], James Schwab[1]

[1]Atmospheric Sciences Research Center, University at Albany, State University of New York, Albany, NY, 12226, United States
[2]Department of Atmospheric and Environmental Sciences, University at Albany, State University of New York, Albany, NY, 12226, United States
[3]School of Engineering and Applied Sciences, Harvard University, Cambridge, MA, 02138, United States
[4]School of Earth and Atmospheric Sciences, Georgia Institute of Technology, Atlanta, GA, 30332, United States
[5]Department of Chemical and Biomolecular Engineering, A. James Clark School of Engineering, University of Maryland, College Park, MD, 20742, United States

*Correspondence to*: Jie Zhang (jzhang35@albany.edu)

**Abstract.** The hygroscopicity of organic aerosol ($\kappa_{OA}$) plays a crucial role in cloud droplet activation and aerosol-radiation interactions. This study investigated the viability of an optical scatter monitor system, featuring two nephelometric monitors (pDR-1500), to determine $\kappa_{OA}$, after knowing the aerosol chemical composition. This system was operated during a mobile lab deployment on Long Island in the summer of 2023, which was executed to coordinate with the Atmospheric Emissions and Reactions Observed from Megacities to Marine Areas (AEROMMA) field campaign. The derived $\kappa_{OA}$ under subsaturated high humidity conditions (RH between 85% and 95%) were categorized based on different aerosol sources, including wildfire aerosol, urban aerosol, and aerosol from rural conditions. The $\kappa_{OA}$ and the OA O:C ratio exhibited linear positive relationships for the urban aerosol and the aerosol from rural conditions, with a much higher slope (0.50 vs. 0.24) for the latter. However, there was no clear relationship between $\kappa_{OA}$ and the OA O/C ratio observed during each period affected by wildfire plumes. The system proposed here could be widely applied alongside the current aerosol component measurement systems, providing valuable insights into the large-scale spatial and temporal variations of OA hygroscopicity.

## 1 Introduction

Aerosol hygroscopic growth under subsaturated high humidity remains one of the most important research topics in aerosol hygroscopicity (Liu et al., 2018; Wang et al., 2022). This phenomenon can directly determine aerosol liquid water (ALW), which can in turn impact the chemical composition and optical properties of aerosols through aqueous reactions and enhanced light scattering under ambient conditions (Ervens et al., 2011). Additionally, it plays a crucial role in the aerosol's ability to form cloud condensation nuclei (CCN), which can significantly influence cloud formation, related indirect

radiative forcing, and in-cloud aqueous chemistry (Seinfeld et al., 2016; Pöhlker et al., 2023). The hygroscopicity parameter under subsaturated conditions ($\kappa_{sub}$, hereafter "$\kappa$" for simplicity) is commonly used to represent the aerosol hygroscopic

activity/growth (Petters et al., 2007). $\kappa$ can be further divided into the inorganic aerosol hygroscopicity ($\kappa_{IOA}$), which can be inferred from aerosol inorganic compound mass concentration, temperature and RH (Lance et al., 2013; Cerully et al., 2015), and organic aerosol hygroscopicity ($\kappa_{OA}$), which is still poorly characterized due to limited knowledge of organic species sources and formation pathways (Jimenez et al., 2009; Shrivastava et al., 2017).

The most common method for deriving $\kappa_{OA}$ involves (1) estimating $\kappa$ from the hygroscopic growth factor (HGF)

measured by the humidified tandem differential mobility analyzers (HTDMA) (Petters et al., 2007; Wu et al., 2013) and (2) calculating $\kappa_{IOA}$ from the inorganic aerosol mass concentration measured by the co-located Aerosol Mass Spectrometer (AMS) (Zhang et al., 2007) or Aerosol Chemical Speciation Monitor (ACSM) (Ng et al., 2011) through the thermodynamic equilibrium model (Fountoukis et al., 2007). However, the combination of these two complicated and expensive instruments (HTDMA and AMS/ACSM) significantly limited their widespread applications for $\kappa_{OA}$ estimation on both spatial and

temporal scales. Numerous studies have reported positive correlations between $\kappa_{OA}$ and the aerosol oxidation state (e.g., O:C ratio) (Chang et al., 2010; Massoli et al., 2010; Cappa et al., 2011; Lambe et al., 2011; Kuwata et al., 2013; Richards et al., 2013) and have suggested a potential method to estimate $\kappa_{OA}$ based on the measured O:C ratio. However, significant discrepancies exist in these relationships, underscoring the critical need for developing a simplified method or system to obtain $\kappa_{OA}$ with the potential for long-term and widespread application, to explore these relationships.

A combination of dry and wet nephelometers has been used to estimate (1) aerosol liquid water content (ALW) (Guo et al., 2015; Kuang et al., 2018) and hygroscopicity (Kuang et al., 2017), relying on the measured aerosol light scattering enhancement factor ($f_{RH}$) (Fierz-Schmidhauser, et al., 2010; Titos, et al., 2016). When combined with aerosol chemical composition data, this approach also allows for the determination of $\kappa_{OA}$ (Kuang et al., 2020; Kuang et al., 2021). These advancements have significantly promoted the application of nephelometers in aerosol hygroscopicity studies, and they also

open up possibilities for using currently very popular, inexpensive optical scatter particle monitors for same purpose (e.g., Thermo pDR-1500, priced around \$5,000; even more affordable options like Purple Air, costing a few hundred dollars, and Plantower PMS series, available for tens of dollars). These inexpensive devices, based on single-wavelength nephelometric technology, could potentially be used to infer aerosol hygroscopicity and associated ALW. However, unlike the commonly dry/wet nephelometers that measure particle scattering coefficients to calculate $f_{RH}$, these inexpensive particle monitors

directly report particle mass concentration as a bulk measurement, essentially functioning as "black boxes". Unfortunately, there are very few studies that explore the potential of these optical particle monitors for such applications.

Zhang et al. (2020) demonstrated the quantitative relationship between the response of the Thermo pDR-1500 (hereafter referred to as "pDR", a type of optical scatter particle monitors, based on single-wavelength nephelometric technology) under subsaturated high relative humidity (RH) conditions and ALW. Building on this, this study extends the application of

the optical scatter instrument system introduced by Zhang et al. (2020) to estimate ALW based on the 2023 summer field measurements. ALW is further used to estimate the $ALW_{OA}$ based on the aerosol chemical composition measured by an

AMS and subsequently used to estimate $\kappa_{OA}$. The derived $\kappa_{OA}$ was categorized based on the different aerosol sources, and its relationship with the measured organic aerosol O/C ratio was discussed. Additionally, a comparison with previous studies is conducted to validate the feasibility of this system.

## 2 Section

### 2.1 Field campaigns

The field measurements were conducted from June 21, 2023 to Sep. 07, 2023 in Long Island, NY, utilizing our Atmospheric Sciences Research Center (hereafter "ASRC") mobile lab. The data collection involved a combination of on-road measurements for some special case days and off-road measurements while parked beside the Flax Pond Marine Laboratory, Stony Brook University. The ASRC mobile lab is a well-equipped platform featuring an aerosol HR-ToF-AMS for aerosol chemical component mass concentration, two pDRs (one for dry aerosol and one for wet aerosol, as described in Fig. 1), a condensation particle counter (CPC) for aerosol number concentration, several gas monitors (i.e., $O_3$, $NO_2$, $CO_2$, HCHO, $CH_4$, etc.) and an Airmar meteorological monitor. Further details about the ASRC mobile lab can be found in Zhang et al. (2018). In this study, the measurements from AMS and the two pDRs were used with a time-averaging period of one hour.

The on-road measurement field campaigns were executed as the "2023 Mobile Laboratory Measurements of the Atmospheric Chemical Evolution in Urban Outflow Plumes and their Interplay with Coastal Meteorology over Long Island" project. This project aims to study the ozone/aerosol chemistry dynamics in the urban plume in the lowest layer under the influence of the coastal meteorology over Long Island, urban heatwave, and other extreme events. It is also designed to fully coordinate with and complement other comprehensive field campaigns - Atmospheric Emissions and Reactions Observed from Megacities to Marine Areas (AEROMMA), the New York City region for the Coastal Urban Plume Dynamics Study (CUPiDS), and the Synergistic TEMPO Air Quality Science (STAQS), during 2023 summer over NYC and its downwind regions including Long Island. More detailed information about the above campaigns can be found at https://csl.noaa.gov/projects/aeromma.

Throughout the measurement period, several periods were significantly influenced by urban plumes from the eastern coastal urban regions, rural plumes from the remote region, or by wildfire plumes transported from western Canada. The days with similar aerosol sources will be classified into one group with a total of three different groups identified in this study. All these provided a unique opportunity to explore the variation of $\kappa_{OA}$ of each group and its relationship with the measured O:C ratio of organic aerosol from each source.

### 2.2 System setup

A schematic of the setup for $\kappa_{OA}$ estimation used in the ASRC mobile lab is depicted in Fig. 1, comprising two pDRs, one Aerodyne high-resolution time-of-flight aerosol mass spectrometer (HR-ToF-AMS, hereafter "AMS"), and one TSI

silica dryer (Diffusion Dryer 3062). During the measurements, one pDR was installed downstream of the silica dryer for the dry aerosol mass concentration (hereafter "pDR$_{dry}$"), and one pDR was directly connected to the ambient air under ambient RH conditions for the wet aerosol mass concentration (hereafter "pDR$_{wet}$"). The AMS was used to measure the non-refractory submicron particles (NR-PM$_1$) chemical component mass concentration (including organic, sulfate, nitrate, ammonia, and chlorine), the O:C ratio, and also used as the reference aerosol mass concentration instrument to calibrate the pDR measurements. Meanwhile, data collected under the lowest relative humidity conditions reported by pDR$_{wet}$ (RH < 45%) were utilized to generate a self-correlation scatterplot between the two pDRs (Fig. S1), which was applied to all data from pDRwet before all further data analysis. During the deployment, the RH in pDR$_{dry}$ ranged between 30% and 45%. We used 45% as the upper RH threshold for self-calibration, based on the following considerations: (1) ISORROPIA II model calculations indicate that aerosol liquid water associated with inorganics (ALW$_{IOA}$) is zero for all data below 45% RH, and (2) submicron internally mixed inorganic-organic particles do not exhibit hygroscopic growth until they reach their deliquescence point, which occurs at approximately 77% RH (Pope et al., 2010; Jing et al., 2016; Bouzidi et al., 2020). In the mobile lab setup, ambient air was drawn at a flow rate of around 56 liters per minute (LPM) into a stainless steel tube with a 2.5 cm diameter, equipped with a PM cyclone (URG-2000-30EC) designed to filter particles larger than 2 μm. The TSI silica dryer and pDRwet were linked to the sampling duct of the stainless steel tube via black conductive tubing with an internal diameter of 4.5 mm. The tubing lengths were approximately 0.3 m for the TSI silica dryer and 1 meter for the pDRwet. After The TSI silica dryer (roughly 0.5 m long), the pDR$_{dry}$ and AMS were connected to the dryer output through a 0.2 m black tubing parallelly. Varied lengths of black tubing were employed to maintain a roughly consistent total airflow path to the pDR$_{dry}$, pDR$_{wet}$, and AMS. The air flow was expected to be turbulent based on the calculated Reynolds Number (RN=30234, as determined from https://www.omnicalculator.com/physics/reynolds-number), and the estimated particle loss of the ambient aerosol, with a size between 100 nm to 1000 nm, from the van inlet to each instrument was less than 1% (https://www.mpic.de/4230607/particle-loss-calculator-plc).

The selection of pDR is based on its capability to report both the temperature and RH of the aerosol flow, along with the aerosol mass concentration. The pDR is a type of nephelometric monitor that utilizes an LED light source with a wavelength of 880nm. It measures particle scattering within a forward scattering angle range of 60 to 80 degrees. The device converts the intensity of the scattered light it detects into mass concentration values based on the factory calibration, which was aligned with a gravimetric standard Arizona Road Dust (Zhang et al., 2018). The calibration factor for the pDR, defined as the ratio of the aerosol mass concentration reported by the pDR to that of a reference instrument, was shown to be directly proportional to the relative scattering intensity calculated using Mie theory (Zhang et al., 2018), based on the lab tests for the mono-disperse particles (90nm, 173nm, 304nm, 490nm, 1030 nm of Polystyrene latex spheres (PSL) particles) and for the poly-disperse particles with four different chemical compositions (NaNO$_3$, (NH$_4$)$_2$SO$_4$, sucrose, and adipic acid). Based on laboratory tests and ambient measurements, the pDR exhibited a unimodal distribution for its calibration factor, peaking around 500 nm. This peak was larger than that of another nephelometric monitor tested in parallel, the TSI DRX (operating at a 660 nm wavelength and 90° scattering angle), which peaked at 300–400 nm. The higher peak for the pDR is attributed to

its use of a longer wavelength. However, the precise value of the calibration factor is further influenced by aerosol composition, which affects the refractive index and, consequently, the relative scattering intensity. These findings raise concerns about the calibration of widely used low-cost particle sensors based on single-wavelength nephelometric technology. Generally speaking, the relative scattering intensity, which will be proportional to the report aerosol mass concentration from these low-cost particle sensors, is influenced by particle size, composition, instruments properties (such as light wavelength and scattering angles), and ambient RH as a factor influencing ALW—an important focus of this study. It is challenging to apply simple calibration factors, derived from laboratory tests on specific aerosol species, to fully correct low-cost sensors. Additionally, the calibration factor for one type of monitor cannot simply be applied to another monitor with different properties (e.g., light wavelength and scattering angles). Addressing these limitations will require further research and targeted calibration efforts specific to each monitor's characteristics.

Furthermore, it was demonstrated that the calibration factor was almost independence to the aerosol wet/dry conditions, and was minimally affected by RH variations within the range of 45 to 95%, maintaining an accuracy with an error margin of less than 5%. This is due to the minimal variation in relative scattering intensity caused by aerosol in this RH range, after considering the influence of ALW. It should be noted that a minimally affected calibration factor means that the ratio of the dry aerosol calibrated mass concentration to the monitor-reported value at 45% RH is very similar to the ratio of the wet aerosol calibrated mass concentration to the monitor-reported value at 95% RH. However, the values for wet aerosol—both the calibrated mass concentration and the monitor-reported value—will be larger than those for dry aerosol due to the presence of aerosol liquid water (ALW) under higher humidity conditions, with further discussed in more detail below using pDR as an example.

In this way, the aerosol mass concentration reported by $pDR_{wet}$ (hereafter "$M_{pDRwet}$", units: $\mu g\ m^{-3}$) can be calibrated based on the calibration factor derived from the aerosol mass concentration measured from $pDR_{dry}$ (hereafter "$M_{pDRdry}$", units: $\mu g\ m^{-3}$) and from the reference instrument (AMS in this study, $M_{AMS}$ for the measured mass concentration, units: $\mu g\ m^{-3}$). Any increase in the mass concentration measured by the calibrated pDRwet compared to that of the calibrated pDRdry can be attributed solely to the presence of ALW (Zhang et al., 2020).

Both pDR devices were fitted with a "Blue Cyclone" and had their flow rates set to 1.5 LPM, achieving an aerosol diameter 50% cut point of 2.5 μm. This cut point was chosen to be 2.5 μm, instead of the 1μm (upper size limit of the AMS), to accommodate the enlargement of aerosols under high RH conditions when using the $pDR_{wet}$. However, the difference in the size range between the pDR devices and the AMS introduced a level of uncertainty to the proposed method, which will be addressed in the following discussion. Aside from the uncertainty due to size differences, the AMS only measures non-refractory aerosols and has limited sensitivity to refractory aerosols (e.g., sea salt), which introduces additional uncertainty and will be discussed further in Section 2.3. It is also important to note that the temperature and RH obtained from $pDR_{wet}$ were measured inside of $pDR_{wet}$ and could be affected by the inside temperature of the mobile lab and the calculated ALW may not accurately represent the real ALW of the ambient aerosol.

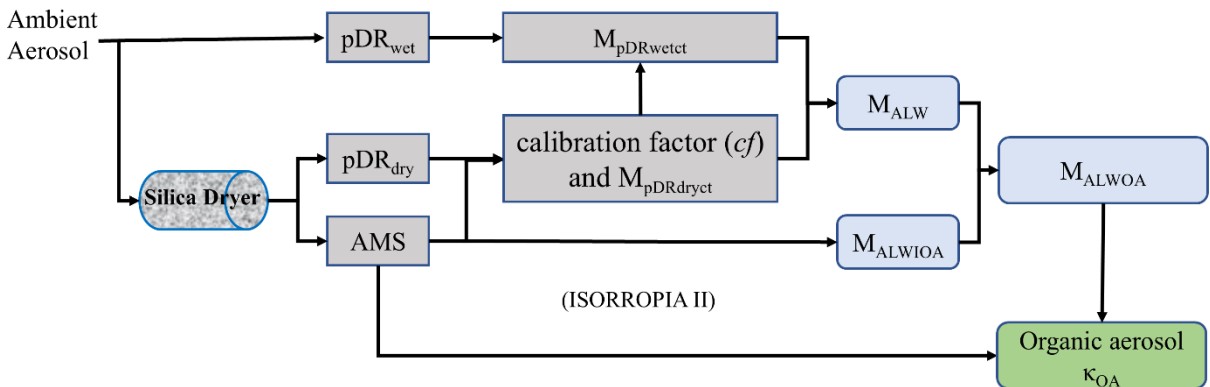

**Figure 1. Schematic of the experimental setup for the organic aerosol hygroscopicity (κ$_{OA}$) estimation.**

As shown in Fig.1 and described more fully in our previous study (Zhang et al., 2020), mass of ALW (hereafter "M$_{ALW}$", units: μg m$^{-3}$) can be obtained from the subtraction of the calibrated aerosol mass concentration from pDR$_{dry}$ (hereafter "M$_{pDRdryc}$") from the calibrated aerosol mass concentration pDR$_{wet}$ (hereafter "M$_{pDRwetc}$"), as shown in Eq.(1):

$$M_{ALW} = M_{PDRwetc} - M_{PDRdryc} \tag{1}$$

Here M$_{pDRdryc}$ was set equal to the aerosol mass concentration measured by AMS (M$_{AMS}$), and a calibration factor (cf = M$_{pDRdry}$/M$_{AMS}$) was applied to M$_{pDRwet}$ to obtain M$_{pDRwetc}$ (M$_{pDRwetc}$= M$_{pDRwet}$/cf). The difference between M$_{pDRwetc}$ and M$_{pDRdryc}$ is attributed to ALW, based on the consideration that the only increasement for the dry aerosol under high RH would be the concentration of the water being absorbed (M$_{ALW}$). Here the cf, obtained from the ratio of M$_{pDRdry}$ to M$_{AMS}$, was applied to determine the calibrated mass concentration of the wet aerosol (M$_{wet}$=M$_{AMS}$+M$_{ALW}$), given that the calibration factor being almost independence to the aerosol wet/dry conditions as described above, as shown in Eq.(2),

$$cf= \frac{M_{pDRdry}}{M_{AMS}} = \frac{M_{pDRwet}}{M_{wet}} \tag{2}$$

Here M$_{wet}$ is the calibrated mass concentration of the wet aerosol (M$_{wet}$=M$_{AMS}$+M$_{ALW}$), and is M$_{pDRwetc}$ in Eq. (1).

The thermodynamic equilibrium model ISORROPIA II (Fountoukis et al., 2007) was used to estimate the ALW taken up by the inorganic aerosol compounds (hereafter "M$_{ALWIOA}$"), based on (1) the inorganic aerosol compound concentrations (NO$_3^-$, SO$_4^{2-}$, NH$_4^+$) measured by AMS with all other metal ions setting 0, and (2) the RH and temperature measured inside of pDR$_{wet}$. The calculated M$_{ALWIOA}$ is then subtracted from M$_{ALW}$ to obtain ALW caused by the organic aerosol compounds (hereafter "M$_{ALWOA}$", as shown in Eq.(3):

$$M_{ALWOA} = M_{ALW} - M_{ALWIOA} \tag{3}$$

Then, the κ$_{OA}$ (units: 1) can be inferred from MALWOA following Eq.(4) (Nguyen et al., 2016):

$$\kappa_{OA} = M_{ALWOA} \div (\rho_w \times \frac{m_{OA}}{\rho_{OA}} \times \frac{RH}{1-RH}) \tag{4}$$

where RH is the relative humidity reported by pDR$_{wet}$, m$_{OA}$ is the AMS measured organic aerosol mass concentration, ρ$_w$ is the water density (1.0 g cm$^{-3}$), and ρ$_{OA}$ is the organic aerosol density. In this study, we used 1.4 g cm$^{-3}$ for ρ$_{OA}$ following the commonly used value (Hallquist et al., 2009; Shakya and Griffin, 2010; Nguyen et al., 2016; Riva et al., 2017; Jiang et al., 2019). However, the ρ$_{OA}$ can vary significantly depending on the sources and formation pathways of organic aerosols, with a range between 1.2 and 1.6 g cm$^{-3}$ based on a recent chamber study (El Mais et al., 2023), introducing some uncertainty into our results. In this study, only the data with RH between [85% 95%] were considered for estimating κ$_{OA}$ in order to (1) match the RH used in HTDMA, (2) reduce the uncertainty of aerosol mass concentration measured by pDR$_{wet}$ under the RH over 95%, which is suggested by the pDR user manual, and also (3) ensure the inorganic aerosol is in an aqueous state. The derived κ$_{OA}$ and ambient temperature and RH can be further used to estimate ambient ALW through the inverse calculations based on Eq.(4) and Eq.(3), and the information of ambient ALW can be very useful for the study of aqueous SOA formations/evolutions. Meanwhile, it also emphasized the possibility of using this system for using direct ambient measurements, very similar to the innovative outdoor dry/wet nephelometer system described by Qiao et al. (2024), without drying aerosols first before analysis as the HDMA (Tang et al., 2019) and without worrying about altering their actual phase state in ambient air (Qiao et al., 2024).

## 2.3 Method uncertainty and limitations

In this study, using AMS as the reference instrument for pDR$_{dry}$ could introduce a certain level of uncertainty for the ALW estimation due to (1) the AMS's limited sensitivity to refractory aerosols (e.g. sea salt), and (2) the discrepancy size range detected by the pDR$_{dry}$ and AMS. The coarse-mode particles (including the coarse-mode refractory aerosols) with diameter between 1 µm and 2.5 µm detected by pDR$_{dry}$ will not be captured by the AMS. The basics assumption here is the chemical composition of the coarse and fine modes is similar to each other throughout the study (Sun et al., 2020), and the ratio of particle water in the fine and coarse modes will equal the ratio of fine and coarse mode dry mass concentration. So that, the estimated M$_{ALW}$ here based on the calibrated aerosol mass concentration from the pDRs using AMS as reference can represent the liquid water in non-refractory PM$_1$. However, significant uncertainty will be introduced in the estimation of κ$_{OA}$, particularly due to the presence of sea salt and other high-κ refractory components in coarse aerosols (AzadiAghdam et al. (2019)), which can greatly increase their hygroscopicity. Due to the limited information on the chemical composition (including refractory components) of fine and coarse aerosols, we can only provide a rough estimate of this uncertainty as a bulk, as shown below. Additionally, this uncertainty was further magnified when calculating M$_{ALWOA}$ based on the estimate M$_{ALWIOA}$ from ISORROPIA II. The absence of measurements for metal ions necessitated the assumption of "0" for all such ions in the ISORROPIA II inputs, further compounding the uncertainty, along with the inherent uncertainties of the ISORROPIA II model itself. Moreover, the uncertainty of calculating κ$_{OA}$ will further come from using the empirical equation Eq.(4) and the assumed value for the density of organic compounds.

To approximate the uncertainty associated with this proposed method, we categorized the measured O:C ratio into bins with an increment of 0.05, ranging from 0.4 to 1.0, for each group with different aerosol source. We then assumed that the

standard deviation of $\kappa_{OA}$ within each bin reflects the uncertainty in the estimated $\kappa OA$, based on the assumption $\kappa_{OA}$ is linearly related to O:C ratio for each specific aerosol source group. The maximum standard deviation of $\kappa_{OA}$ across all bins of the identified three groups was determined to be 0.08 with the mean value of $\kappa_{OA}$ for this bin as 0.18, which was expected as the upper limit of the uncertainty for $\kappa_{OA}$. More detailed information of the distribution of $\kappa_{OA}$ in each bin for each group with different aerosol sources and its relationship with the measured O:C ratio is discussed in the following section "Variation of $\kappa_{OA}$ with different aerosol sources". Meanwhile, it's crucial to acknowledge that this study does not account for the impact of black/brown carbon on the results, as both the pDR devices and the AMS do not detect black/brown carbon.

It is important to note that the derived $\kappa_{OA}$ values in this study were not continuous, as we could only obtain them under high relative humidity (RH) conditions (85% to 90%). Additionally, our current inability to maintain aerosol under such high RH conditions limited the laboratory calibration and verification of this method using substances with known hygroscopic parameters (Fierz-Schmidhauser, et al., 2010; Zieger et al, 2013; Han et al., 2022), even though this method is theoretically feasible. To resolve this issue, one possible update of this system could be adding a humidifier system to the pDR to get wet aerosol with RH between 85% to 95%, and the possible set-up for humidifier system could include a Perma Pure MH-series humidifier, water pumps and tanks (red dash box in Figure 2). This will make this system more be similar to the widely used humidified nephelometer system (Guo, et al., 2015; Burgos et al., 2019, Fierz-Schmidhauser et al., 2010; Kuang et al., 2017,2018,2020, 2021).

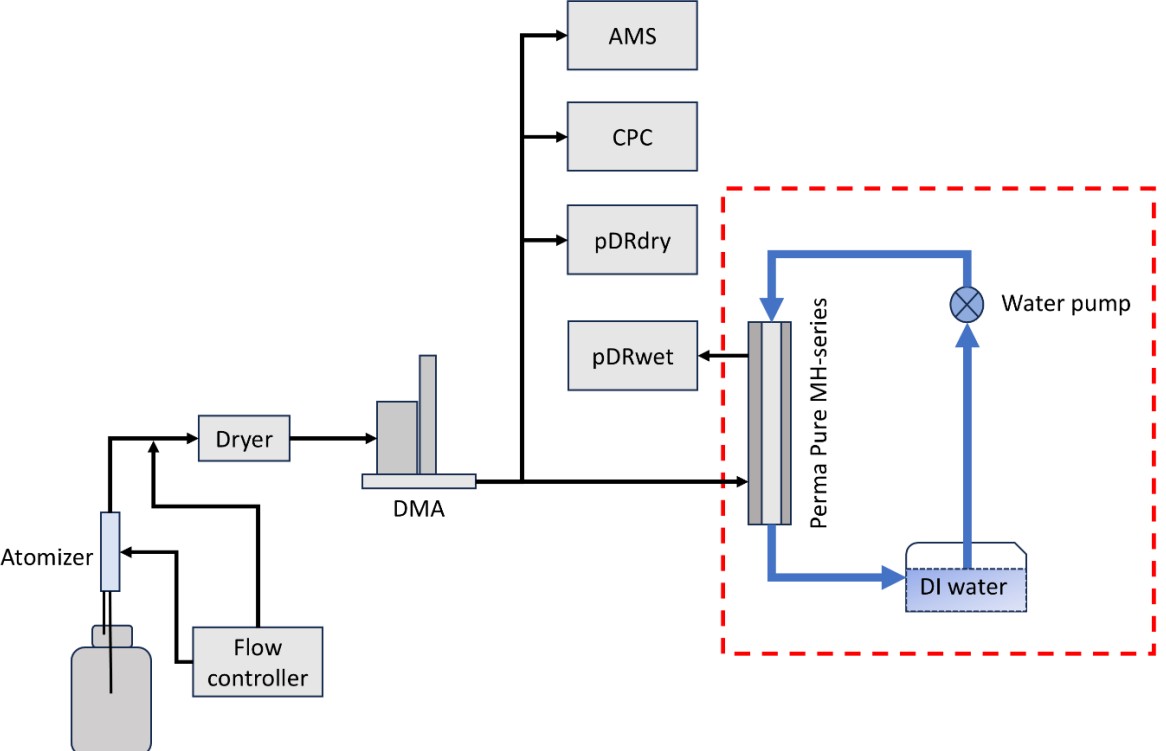

**Figure 2. proposed instrument setup for lab calibration**

The proposed instrument setup for lab calibration will include an atomizer to produce aerosol, which will be dried through the dryer. Then, the Differential Mobility Analyzer (DMA) will pick up different size of aerosols, with one of branch aerosol flow to the humidifier system to get wet and then to pDRwet, and another three branches of aerosol flows to pDRdry, AMS, and condensation particle counter (CPC). The substances/aerosol for testing will include the organic aerosols with known hygroscopic parameter (Han et al., 2022), the inorganic aerosols (i.e., $(NH_4)_2SO_4$, Fierz-Schmidhauser et al.,

2010), as well as their mixing solutions of organic and inorganic. Due to limited resources, this proposed instrument setup is not feasible at this moment, and the lab calibration is not included in this study. However, we hope this will inspire other research groups with this set-up to conduct these lab tests to better quantify the uncertainty of this method for pDRs. Given that the pDR is a type of single-wavelength nephelometric monitor, it's logical to consider that other brands commonly used low-cost nephelometric monitors (e.g., Purple Air, Plantower PMS series) might offer similar capabilities, and related lab

test would be also highly recommended. 3 Results and discussion

### 3.1 Overview of measurements

The time series of all calibrated aerosol mass concentrations measured by the pDRs are shown in Fig. 3a, revealing significant discrepancies between pDRdryc and pDRwetc under high RH conditions (Fig. 3b). This highlights the contribution of ALW to the response of the pDRwetc. As shown in Fig. 3b, the mass growth factor ($=M_{pDRwetc}/M_{pDRdryc}$) was mainly

between 2 to 4 under RH range of 90% to 100%, with an averaged value of 2.5, which was generally higher than the value under the RH range of 80% to 90% with an averaged value of 1.3. Notably, there were several points with growth factors around 1.3 under their RH range [90% 100%], suggesting their weaker hygroscopicity compared to the points with a growth factor between 2 and 4. This discrepancy also implies different sources for these two distinct RH ranges.

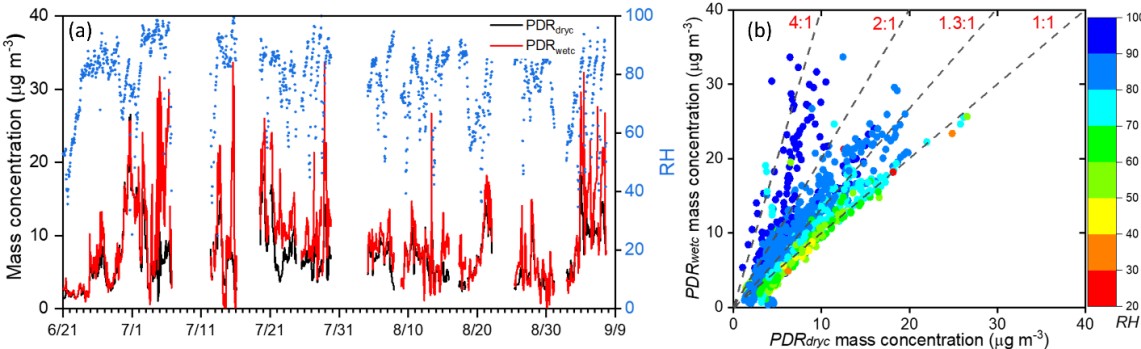

**Figure 3. (a) The time series of 1-h average aerosol mass concentration measured by pDRwetc and pDRdryc, and (b) the correlation scatter plot of pDRwetc and pDRdryc colored by RH. The dashed lines represent the ratio lines of $M_{pDRwetc}$ to $M_{pDRdryc}$ at 1:1, 1.3:1, 2:1, and 4:1.**

As described in the "METHODS" section, only the pDRwet with RH between [85% 95%] was considered for κ

estimation, and the calculated $M_{ALWIOA}$, $M_{ALWOA,}$ and $M_{ALW}$ ($M_{ALW} = M_{ALWIOA} + M_{ALWOA}$) are shown in Fig. 4. ALW could be

as high as about 18.6 µg m$^{-3}$ with the related mass growth factor around 2.9. During the initial half of the deployment, there were some points with $M_{ALW_{OA}}$ below 0, and this occurred when $M_{ALW_{IOA}}$, as estimated from the ISORROPIA II model, exceeded the total $M_{ALW}$ derived from the two pDR measurements. Such negative values can be attributed to previously discussed uncertainties in either the calibration of the pDR devices or the estimations made by the ISORROPIA II model,

and will also result negative $\kappa_{OA}$, as described below. When considering all the points with $M_{ALW_{OA}}$ over 0, $ALW_{OA}$ showed significant contributions to the total wet aerosol mass concentration with an average fraction of 27% and a range of 15% to 39% within the 25th to 75th percentiles of the dataset.  This underscores the necessity of obtaining accurate $\kappa_{OA}$ values to better obtain $ALW_{OA}$ and evaluate its impact on aerosol evolution.

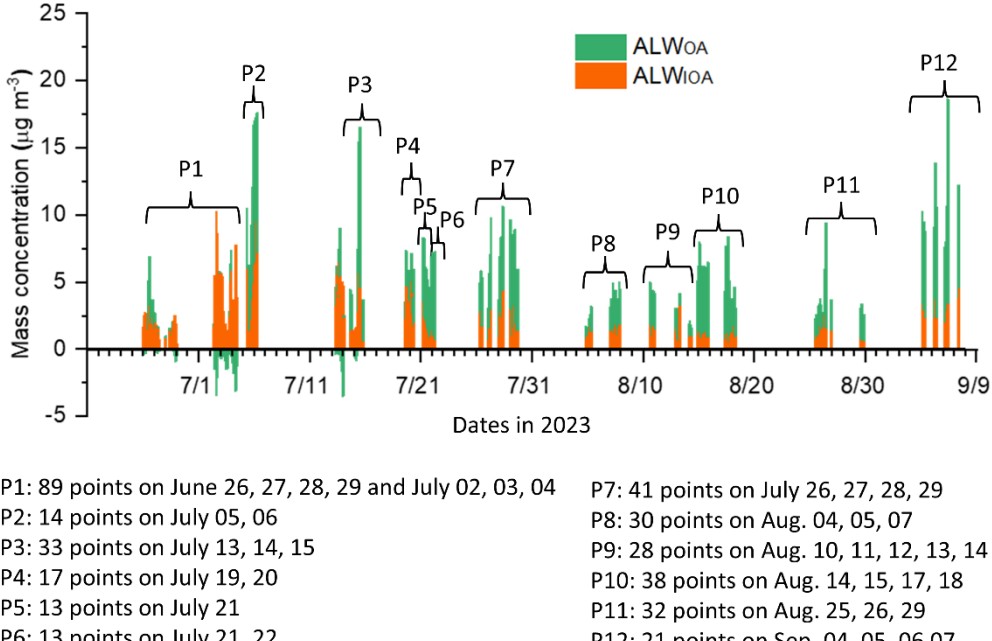

P1: 89 points on June 26, 27, 28, 29 and July 02, 03, 04     P7: 41 points on July 26, 27, 28, 29
P2: 14 points on July 05, 06                                                   P8: 30 points on Aug. 04, 05, 07
P3: 33 points on July 13, 14, 15                                             P9: 28 points on Aug. 10, 11, 12, 13, 14
P4: 17 points on July 19, 20                                                   P10: 38 points on Aug. 14, 15, 17, 18
P5: 13 points on July 21                                                         P11: 32 points on Aug. 25, 26, 29
P6: 13 points on July 21, 22                                                   P12: 21 points on Sep. 04, 05, 06,07

**Figure 4. The time series of stacked column of the $ALW_{OA}$ and $ALW_{IOA}$ with a time resolution of 1-hr. (P1-P12 denote the different sub-periods mentioned in the following text with the data points and time periods of each subperiod indicated.)**

### 3.2 Variation of κOA with different aerosol sources

The box and whiskers distribution of the derived $\kappa_{OA}$ based on Eq. (4) for each sub-period is shown in Fig. 5a, alongside the HR-ToF-AMS measured PM$_1$ mass concentration in Fig. 5b. Sub-periods were categorized based on the back trajectories of each subperiod (Fig. S2-S7).They were divided into three groups with different aerosol sources based on their similar back trajectories, mass concentration, as well as $\kappa_{OA}$, including one group with aerosol having urban sources (marked in grey in Fig. 5, hereafter "Group1(urban)", Fig. S2 for their back trajectories), one group with aerosol having rural sources

(marked in green in Fig. 5, hereafter "Group2(rural)", Fig. S3 for their back trajectories), and one group with aerosol affected

by the wildfire plumes (marked in red in Fig. 4, hereafter "Group3(wildfire)", Fig. S4-S7 for their back trajectories). Generally speaking, Group1(urban) showed relatively higher mass concentration and lower $\kappa_{OA}$, which agrees with findings from previous studies that the urban aerosol generally has a low hygroscopicity activity (Wu et al., 2016; Hong et al., 2018). At the same time, the points with $\kappa_{OA}$ below 0 were predominantly found in P1 of Group1(urban) with values dipping to as low as -0.08, and these negative values fell within the expected upper limited uncertainty of $\kappa_{OA}$ of 0.08 given the averaged $\kappa_{OA}$ of P1 being around 0. The relatively low $\kappa_{OA}$ in P1 followed their low O:C ratio, as shown in Fig. S8. Conversely, Group2(rural) exhibited higher values of $\kappa_{OA}$ compared to other groups, which can be attributed to their exposure to long-term transport/reactions and consequently, a stronger hygroscopicity activity. Meanwhile, the data points of the subperiods (P10 and P11) of Group2(rural) were highly scattered, especially for plume back trajectories over the ocean, highlighting the uncertainty caused by the marine sea salt aerosol. The Group3(wildfire)demonstrated a big range of $\kappa_{OA}$, with the subperiod "P4" having the lowest $\kappa_{OA}$ (an averaged value of 0.02, near to hydrophobic organics, Kuang et al., 2020; Han et al., 2022) and the highest mass concentration. Additionally, the subperiod "P4" exhibited the most notable transport pathway from western Canada to NYC metro regions (Fig. S4) compared to other wildfire plume cases (Fig. S5-S7). Considering all four of these cases of wildfire aerosol having an original wildfire source in western Canada, it is reasonable to infer the wildfire $\kappa_{OA}$ could be strongly affected by the burning time of the original forests, the related burning conditions (i.e., smoldering vs. flaming, etc.), the transport time from west to east, etc. (Garofalo et al., 2019), resulting in significant variation between different cases, warranting further investigation.

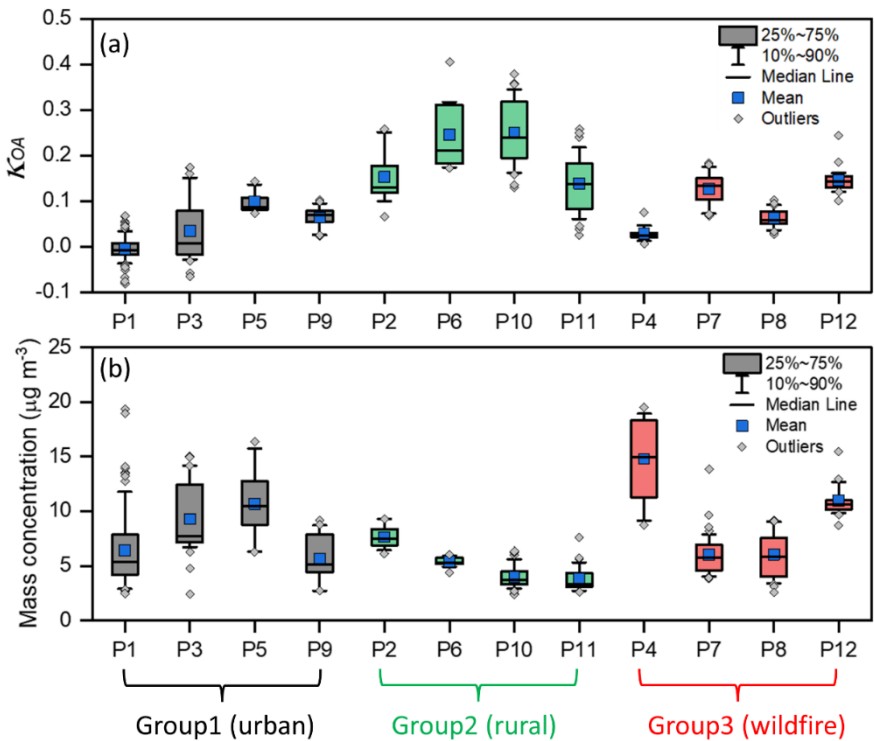

**Figure 5. The box and whiskers distribution of κOA and aerosol total mass concentration for each subperiod. (The time resolution of each data point is 1 hr. The subperiods being affected by urban plumes are marked in grey and categorized as Group1(urban), the ones being affected by rural environments are marked in green and categorized as Group2(rural), and the ones being affected by wildfire plumes are marked in red and categorized as Group3(wildfire)).**


The derived subsaturated hygroscopicity of organic compounds in both Group1(urban) and the Group2(rural) exhibited a tight relationship with their O:C ratio, with the $\kappa_{OA}$ increasing as the O:C level rose while distinct slopes for each group, as shown in Fig. 6a and Fig. S8. The urban aerosol showed a much smaller linear slope (~0.24) between $\kappa_{OA}$ and O:C compared to the rural aerosol, which had a steeper linear slope of 0.50. The fitted linear slopes of this study closely resembled previous

studies having similar organic aerosol sources. This supports previous findings that the hygroscopicity of urban organic aerosols is much less sensitive to variation in their oxidation level than rural organic aerosols (Wu et al., 2016; Hong et al., 2018). Fig. 6a also presents the derived slope from previous studies for various atmospheric conditions using more precise instruments for $\kappa_{OA}$, with the HTDMA for the urban aerosol in China by Hong et al. (2018) and the forest aerosols in Japan by Deng et al. (2019), the CCN counter (CCNc) for the rural mountain aerosols in USA by Zhang et al. (2019). The slope of

0.24 of Group1(urban) was near to the value reported in Guangzhou, China by Hong et al. (2018) and the slope of 0.50 aligned with findings from the forest/mountain aerosols (Deng et al., 2019; Zhang et al., 2019). The close alignment between the results of this study and those from previous research underscores the viability of this simpler system to offer reasonable estimates of $\kappa_{OA}$ in comparison to more precise and costly instruments, such as the HTDMA and the CCNc. Meanwhile, the

near-constant trends of $\kappa_{OA}$ are showed for each period affected by the wildfire plumes (Fig. 6b), and that there were no clear

linear relationships between $\kappa_{OA}$ and O:C for each period affected by the wildfire. This could be related to the complexity of the wildfire plumes and their long-term transport from west to east. More specially, they showed a negative relationship when combining all four wildfire periods (Fig. 6b), and further studies will need to verify this and investigate the possible reasons.

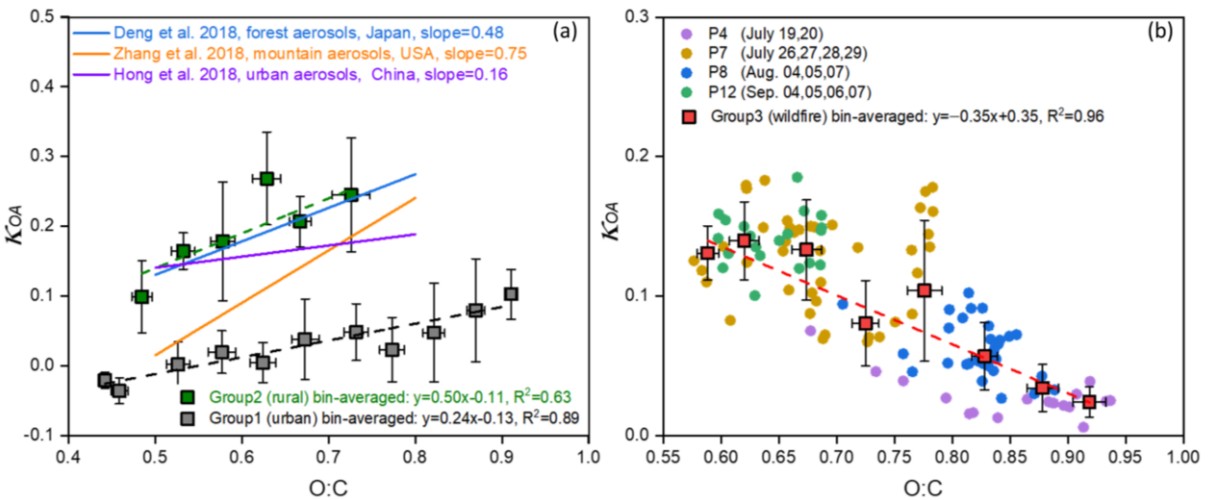

**Figure 6. The relationship between $\kappa_{OA}$ and O:C of each group. (a) The relationship between $\kappa_{OA}$ and O:C for Group1(urban) (marked by grey) and Group2(rural) (marked by green). (Data points located in each O:C bin was averaged to obtain the bin-averaged $\kappa_{OA}$ and O:C, with the error bar showed their standard deviation in each bin. The bin-averaged $\kappa_{OA}$ and O:C were fitted using the linear regression fit with the fitting line in dash. Meanwhile, the fitted slopes between $\kappa_{OA}$ vs. O:C from previous studies were presented in solid lines, and were used to compare with**
**the results of this study and to verify the feasibility of the proposed method). (b) The relationship between bin-averaged $\kappa_{OA}$ and O:C for the aerosols affected by the wildfire transports (in red square with their standard deviation as error bar), and the relationship between $\kappa_{OA}$ and O:C of each subperiod of Group3(wildfire) with a time resolution of 1-hr.**

Once again, the distinctly different relationships between $\kappa_{OA}$ and O:C between these three groups of organic aerosols indicate the substantial uncertainty in describing the hygroscopicity using a simplified average O:C ratio without considering the possible organic aerosol sources (Kuang et al., 2020; Han et al., 2022), and also highlights the necessity of deriving $\kappa_{OA}$ based on direct measurements.

## 4 Conclusions

An inexpensive single-wavelength nephelometers system, containing two pDRs with one for dry aerosol and one for wet aerosol, was used to derive the organic aerosol hygroscopicity parameter ($\kappa_{OA}$) under subsaturated high humidity conditions (RH between [85% 95%]), after knowing the aerosol chemical compound mass concentrations. The derived $\kappa_{OA}$ for the measurement period was largely dependent on the aerosol sources and showed different relationships with the organic aerosol oxidant level (i.e., O:C ratio in this study) for each classification of the aerosol source. $\kappa_{OA}$ showed a positive linear

relationship with O:C ratio for the urban aerosol and the rural aerosol with a much higher slope for the latter (0.24 urban vs. 0.50 rural). Meanwhile, the magnitude of $\kappa_{OA}$ of rural aerosol is much higher than the value of urban aerosol. The fitted relationships agreed well with previous studies, supporting the feasibility of this simple system to estimate $\kappa_{OA}$. No clear relationship was shown for each period when the organic aerosol was influenced by the transported wildfire plumes. These large different $\kappa_{OA}$ vs. O:C relationships, including both slopes and magnitudes, for each group imply the necessity of estimation of $\kappa_{OA}$ through direct measurements, rather than through a simple dependent relationship based on one kind of aerosol other properties (i.e., O:C ratio).

This approach offers a cost-effective alternative (given that two pDRs cost around $10,000) for estimating the $\kappa_{OA}$ of ambient aerosols during field campaigns, especially when utilizing AMS or ACSM to measure the mass concentration of aerosol chemical compounds in situations where like HTDMA or the dry/wet conventional nephelometers system are not available. Another possible more broadly application of this system could be to the US EPA Chemical Speciation Network (CSN) network for the period averaged $\kappa_{OA}$ after knowing the time averaged mass concentration of each chemical compounds. Further studies for this method on the even more affordable options like Purple Air and Plantower PMS series are warrantied, and results from these further investigations would largely enhance the role of this kind of optical particle monitors in aerosol hygroscopicity studies, especially given the growing popularity of these monitors in community disadvantage studies, where most projects rely on such devices to monitor $PM_{2.5}$. The potential widespread use of this method is expected to enhance our understanding of $\kappa_{OA}$ variations and their influence on CCN activities across various spatial and temporal scales. Moreover, it enables the calculation of ambient ALW from the derived $\kappa_{OA}$, taking into account ambient temperature and RH, which is particularly valuable for studies on atmospheric aqueous phases and the formation of secondary organic aerosols.

*Data availability.* The data set is available upon request from the corresponding author.

*Author contributions.* JZ preforms the calculation and data analysis; TZ and AC helped to the data collection; YL, MS, PL, AA, JS helped to interpret the results and revised the manuscript. JZ wrote the paper with contributions from all coauthors.

*Competing interests.* The author has declared that there are no competing interests.

*Acknowledgements.* This work has been supported through the New York State Energy Research and Development Authority (NYSERDA) Contract 183868. We acknowledge the support and assistance of NYS DEC personnel, Stony Brook University, and, in particular, the curator of the Flax Pond Marine Laboratory, Mr. Stephen Abrams.

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
