# Peer review of "Technical note: Quantified organic aerosol subsaturated hygroscopicity by a simple optical scatter monitor system through field measurements"

_EGUsphere, 2024_

## Author Comment (AC1)

**Reply to Referee #2**

**General comments:**

This manuscript presents and discusses results obtained via a relatively low-cost method for quantifying the hygroscopicity of organic aerosol. Two pDRs (a single wavelength nephelometer-like instrument) are operated along with an AMS to estimate the mass of water taken up by submicron organic aerosol particles. By assuming a density of the organics (among other assumptions), the hygroscopicity of the organics (K_OA) is estimated. K_OA is evaluated and discussed in the context of O:C ratio as well as for air masses of different origin.

Overall the paper is easy to follow and concise. The analysis is enjoyable to read. It is suitable for publication in ACP, and there are just a handful of assumptions that I would like to see discussed and/or explained in greater detail.

**R: We sincerely thank the reviewer for his/her thorough and thoughtful comments that helped improve our manuscript substantially. We have read all comments carefully and have responded to them each in turn. Below are our point-by-point responses to reviewer #2 comments. Author responses are in bold black. Modifications to the manuscript are in normal front. Line numbers in the response correspond to those in the revised manuscript text file (tracked version).**

Specific comments:

**#1.** Line 91 - I was a bit surprised to see 45% RH used as a "dry" humidity where the pDR_dry and pDR_wet were compared (well, really just < 45% conditions... but RH is not measured below this value). I know of a few organic acids that do not display deliquescence and gradually take up water with increasing RH (e.g., Pope et al., 2010), and there are likely other organics I am not aware of. Do you have a reason to believe <45% RH was sufficient for the comparison you were after? Was the composition mostly dominated by inorganic species that would deliquesce at RH values > 45%? Perhaps you might want to discuss this possibility for organics to take up a certain amount of water with increasing RH and how that may affect this comparison between the pDRs.

**R: Thank you so much for this comment. The RH in pDR$_{dry}$ during the deployment is generally between 30% to 45%, and we used 45% as the upper RH point for the self-calibration in this modified version with the ISORROPIA II calculation showing all these points with RH below 45% with an ALWIOA being 0. Meanwhile, previous HTDMA studies indicate that submicron, internally mixed inorganic-organic particles do not exhibit hygroscopic growth until they reach their deliquescence point at approximately 77% RH (Jing et al., 2016; Bouzidi et al., 2020; Pope et al., 2010). Therefore, we consider 45% to be a safe value for conducting self-calibration comparisons between pDR$_{dry}$ and pDR$_{wet}$. We updated the text, as** "During the deployment, the RH in pDR$_{dry}$ ranged between 30% and 45%.

We used 45% as the upper RH threshold for self-calibration, based on the following considerations: (1) ISORROPIA II model calculations indicate that aerosol liquid water associated with inorganics (ALW$_{IOA}$) is zero for all data below 45% RH, and (2) submicron internally mixed inorganic-organic particles do not exhibit hygroscopic growth until they reach their deliquescence point, which occurs at approximately 77% RH (Pope et al., 2010; Jing et al., 2016; Bouzidi et al., 2020)." **(Line 105-109)**

**#2.** Line 101 - I'm unfamiliar with the pDR and would appreciate a bit more discussion. I see that it uses 880 nm as the wavelength. That's pretty high considering that you are interested in using it to study the hygroscopicity of submicron particles, right? Furthermore, it seems that the instrument was originally calibrated with Arizona Road Dust, which sounds like it would consist of relatively large particles (mostly > 1um?) , although that's just a guess/assumption I'm making. Anyway, due to the larger wavelength it would be great if you could briefly discuss the challenges and/or any previously determined competence of the pDR for analyzing submicron particles. Have results from the pDR ever been compared to those from a nephelometer operating at a comparable wavelengthg and size range of particles?

**R: Thank you so much for this comment. We added more information about pDR based on our previously studies, as well as our concerns/suggestions for using such kinds of nephelometer, as** "The calibration factor for the pDR, defined as the ratio of the aerosol mass concentration reported by the pDR to that of a reference instrument, was shown to be directly proportional to the relative scattering intensity calculated using Mie theory (Zhang et al., 2018), based on the lab tests for the mono-disperse particles (90nm, 173nm, 304nm, 490nm, 1030 nm of Polystyrene latex spheres (PSL) particles) and for the poly-disperse particles with four different chemical compositions (NaNO$_3$, (NH$_4$)$_2$SO$_4$, sucrose, and adipic acid). Based on laboratory tests and ambient measurements, the pDR exhibited a unimodal distribution for its calibration factor, peaking around 500 nm. This peak was larger than that of another nephelometric monitor tested in parallel, the TSI DRX (operating at a 660 nm wavelength and 90° scattering angle), which peaked at 300–400 nm. The higher peak for the pDR is attributed to its use of a longer wavelength. However, the precise value of the calibration factor is further influenced by aerosol composition, which affects the refractive index and, consequently, the relative scattering intensity. These findings raise concerns about the calibration of widely used low-cost particle sensors based on single-wavelength nephelometric technology. Generally speaking, the relative scattering intensity, which will be proportional to the report aerosol mass concentration from these low-cost particle sensors, is influenced by particle size, composition, instruments properties (such as light wavelength and scattering angles), and ambient RH as a factor influencing ALW— an important focus of this study. It is challenging to apply simple calibration factors, derived from laboratory tests on specific aerosol species, to fully correct low-cost sensors. Additionally, the calibration factor for one type of monitor cannot simply be applied to another monitor with different properties (e.g., light wavelength and scattering angles). Addressing these limitations

will require further research and targeted calibration efforts specific to each monitor's characteristics." **(Line 124-141)**

**#3.** Fig. 1: Small thing, but it says "calibrate factor" in the figure, whereas in the text it is always referred to as "calibration factor." I suggest being consistent if you can!

**R: Thank you so much for this comment. We corrected it to calibration factor, as:**

[Figure]

**#4.** Line 127 - I am trying to wrap my brain around the application of the fine mode calibration factor (determined by the ratio of masses measured by the pDR_dry and AMS) to the dry and wet pDR mass concentrations. Maybe a bit more discussion would help me (and potentially others). The calibration factor is based on a difference in dry aerosol mass between the pDR and the AMS, which will most likely have a density > 1 g cm-3 and refractive index greater than 1.33. However, it sounds like much of the mass measured by the pDR_wet can come from water, which has a lower density (1 g cm-3) and refractive index (1.33) than most dry aerosol components. I'm curious if the calibration factor is robust in situations when much of the mass measured by the pDR_wet is from water? I understand you need some way to correct pDR_wet, but maybe you could discuss how the calibration factor is based on dry aerosol and may not apply perfectly when needing to correct a mass measurement (from pDR_wet) that is potentially largely composed of water. And if it is not as big of a deal as I am making it please help me understand why, thanks.

**R: Thank you so much for this comment, and sorry for the unclear caused by the incomplete information provided in the previous version. We added more information to describe this, as:**

"Furthermore, it was demonstrated that the calibration factor was almost independence to the aerosol wet/dry conditions, and was minimally affected by RH variations within the range of 45 to 95%, maintaining an accuracy with an error margin of less than 5%. This is due to the minimal variation in relative scattering intensity caused by aerosol in this RH range, after considering the

influence of ALW. It should be noted that a minimally affected calibration factor means that the ratio of the dry aerosol calibrated mass concentration to the monitor-reported value at 45% RH is very similar to the ratio of the wet aerosol calibrated mass concentration to the monitor-reported value at 95% RH. However, the values for wet aerosol—both the calibrated mass concentration and the monitor-reported value—will be larger than those for dry aerosol due to the presence of aerosol liquid water (ALW) under higher humidity conditions, with further discussed in more detail below using pDR as an example. " **(Line 142-150)**

**And**

"The difference between $M_{pDRwetc}$ and $M_{pDRdryc}$ is attributed to ALW, based on the consideration that the only increasement for the dry aerosol under high RH would be the concentration of the water being absorbed ($M_{ALW}$). Here the cf, obtained from the ratio of $M_{pDRdry}$ to $M_{AMS}$, was applied to determine the calibrated mass concentration of the wet aerosol ($M_{wet}=M_{AMS}+M_{ALW}$), given that the calibration factor being almost independence to the aerosol wet/dry conditions as described above, as shown in Eq.(2),

$$cf = \frac{M_{pDRdry}}{M_{AMS}} = \frac{M_{pDRwet}}{M_{wet}} \qquad (2)$$

Here $M_{wet}$ is the calibrated mass concentration of the wet aerosol ($M_{wet}=M_{AMS}+M_{ALW}$), and is $M_{pDRwetc}$ in Eq. (1). " **(Line 174-181)**

**#5.** Line 136 - I see that you assume a single density for the organic species. I understand you have to assume a density because you do not have size-resolved information about your submicron organic aerosols. As you showed in your study, properties of organic aerosol can be largely different depending on the source, and I'm sure organic densities are subject to this variation (don't have citations off the top of my head, but I'm sure they are out there). Perhaps you can at least mention the organic density likely spanned a range and explain why you chose 1.4 g cm$^{-3}$ (i.e., at least provide a citation as I don't believe there is one now).

**R: Thank you so much for this comment. We added more discussion about this, as "**In this study, we used 1.4 g cm$^{-3}$ for $\rho_{OA}$ following the commonly used value (Hallquist et al., 2009; Shakya and Griffin, 2010; Nguyen et al., 2016; Riva et al., 2017; Jiang et al., 2019). However, the $\rho_{OA}$ can vary significantly depending on the sources and formation pathways of organic aerosols, with a range between 1.2 and 1.6 g cm$^{-3}$ based on a recent chamber study (El Mais et al., 2023), introducing some uncertainty into our results.**" (Line 191-195)**

**#6.** Line 149-152 - I was a little confused by this stated assumption: "By simply assuming a constant mass ratio for the chemical composition of fine-mode and coarse-mode particles, the

ratio of MALW associated with fine-mode particles to that associated with coarse-mode particles will correspond to the dry aerosol mass concentration of each mode."

Perhaps I am not following, but I thought this study was focused solely on submicron mass because the pDR_dry mass is corrected to be equal that measured by the AMS, which is certainly submicron. And then pDR_wet is also corrected by that same correction factor, although the pDR_wet understandably samples up to 2.5 microns to be able to capture the hygroscopic growth of dry submicron particles (sampled by the pDR_dry and AMS). So how would the coarse mode be represented in this study? I was thinking it got "corrected out" by aligning everything to the AMS mass...

Second, this assumption makes me nervous: "the ratio of MALW associated with fine-mode particles to that associated with coarse-mode particles will correspond to the dry aerosol mass concentration of each mode."

I would expect the fine and coarse mode to take up water differently, largely because they are typically composed of different chemical components. For example, see Fig. 10B in AzadiAghdam et al. (2019), where derived kappa values are not consistent with particle size and are largely sensitive to the presence of sea salt and certain inorganic species, which are typically found in varying amounts between the fine and coarse mode.

That all being said, can you provide more insight into why you believe the assumption you are making is a good one? What is it founded upon? Can you provide a reference or justification? Thanks.

**R: Thank you so much for this comment. We rewrote it to make it more clear, as well as highlighting the large uncertainty introduced, as "**The basics assumption here is the chemical composition of the coarse and fine modes is similar to each other throughout the study (Sun et al., 2020), and the ratio of particle water in the fine and coarse modes will equal the ratio of fine and coarse mode dry mass concentration. So that, the estimated $M_{ALW}$ here based on the calibrated aerosol mass concentration from the pDRs using AMS as reference can represent the liquid water in non-refractory $PM_1$. However, significant uncertainty will be introduced in the estimation of $\kappa_{OA}$, particularly due to the presence of sea salt and other high-$\kappa$ refractory components in coarse aerosols (AzadiAghdam et al. (2019)), which can greatly increase their hygroscopicity. Due to the limited information on the chemical composition (including refractory components) of fine and coarse aerosols, we can only provide a rough estimate of this uncertainty as a bulk, as shown below.**" (Line 208-216)**

**7. Line 189 - Small thing, but it would be clearer for me if you explicitly wrote "percentile" with the [25%-75%] or some other explanation for what this range is referring to.**

**R: Thank you so much for this comment. We rewrote it, as "**a range of 15% to 39% within the 25th to 75th percentiles of the dataset.**" (Line 280-281)**

**#8.** Line 196-197 - I was curious about your categorization of the different HYPSLIT back-trajectories. Was this totally subjective or was there more of a process to it? I ask since the lowest-level back-trajectory (red line) on 06 July 2023 in Fig. S3 appears to stay near large metropolitan areas for much of the time before eventually moving out into a more (relatively) rural part of New York. I'm just curious if back-trajectories heading in the northwest direction (or from over the Atlantic Ocean as in panel for 25 August 2023) from the measurement site were considered rural no matter what or if there were more criteria to the classifications?

**R: Thank you so much for this comment. We sperate these three different groups based on the combination consideration of their back trajectories, mass concentration, as well as $\kappa_{OA}$, and updated the text, as "**They were divided into three groups with different aerosol sources based on their similar back trajectories, mass concentration, as well as $\kappa_{OA}$**" (Line 293-294)**

**#9.** Line 279 - The fact that you were not able to calibrate with particles of known composition (and therefore known kappa values) seems like a big deal to me and significant limitation to the study, especially as you are trying to present an alternative and lower-cost method to what has been often used in the past. I find it interesting that the inability to calibrate was not mentioned until the Conclusions section. I would strongly suggest moving this information to the Methods section where you previously were discussing the various limitations associated with the proposed method (the paragraphs right before the Results and Discussion). I appreciate that you propose an alternative instrument set-up where the lab calibration would be possible.

**R: Really thank you so much for this comment. We proposed a potential design for this type of test, but it could not be realized in our lab at this time due to limited resources. Following your suggestion, we moved the design to the methods section, where we describe the instrument setup. We hope this will inspire other research groups to conduct lab tests to better quantify the uncertainty of this method for pDRs, as well as for other commonly used low-cost nephelometric monitors (e.g., Purple Air, Plantower PMS series, etc.) which will provide new insights for their applications. We added related information in the text, as "**It is important to note that the derived $\kappa_{OA}$ values in this study were not continuous, as we could only obtain them under high relative humidity (RH) conditions (85% to 90%). Additionally, our current inability to maintain aerosol under such high RH conditions limited the laboratory calibration and verification of this method using substances with known hygroscopic parameters (Fierz-Schmidhauser, et al., 2010; Zieger et al, 2013; Han et al., 2022), even though this method is theoretically feasible. To resolve this issue, one possible update of this system

could be adding a humidifier system to the pDR to get wet aerosol with RH between 85% to 95%, and the possible set-up for humidifier system could include a Perma Pure MH-series humidifier, water pumps and tanks (red dash box in Figure 2). This will make this system more be similar to the widely used humidified nephelometer system (Guo, et al., 2015; Burgos et al., 2019, Fierz-Schmidhauser et al., 2010; Kuang et al., 2017,2018,2020, 2021).

[Figure]

**Figure 2. proposed instrument setup for lab calibration**

The proposed instrument setup for lab calibration will include an atomizer to produce aerosol, which will be dried through the dryer. Then, the Differential Mobility Analyzer (DMA) will pick up different size of aerosols, with one of branch aerosol flow to the humidifier system to get wet and then to pDRwet, and another three branches of aerosol flows to pDR$_{dry}$, AMS, and condensation particle counter (CPC). The substances/aerosol for testing will include the organic aerosols with known hygroscopic parameter (Han et al., 2022), the inorganic aerosols (i.e., $(NH_4)_2SO_4$, Fierz-Schmidhauser et al., 2010), as well as their mixing solutions of organic and inorganic. Due to limited resources, this proposed instrument setup is not feasible at this moment, and the lab calibration is not included in this study. However, we hope this will inspire other research groups with this set-up to conduct these lab tests to better quantify the uncertainty of this method for pDRs. Given that the pDR is a type of single-wavelength nephelometric monitor, it's logical to consider that other brands commonly used low-cost nephelometric monitors (e.g., Purple Air, Plantower PMS series) might offer similar capabilities, and related lab test would be also highly recommended.**" (Line 236-257)**

**#9.** Also is "varication" supposed to be "verification" in Line 279?

**R: Thank you for this comment. We corrected it, and moved it to new section 2.3, as "**It is important to note that the derived $\kappa_{OA}$ values in this study were not continuous, as we could only obtain them under high relative humidity (RH) conditions (85% to 90%). Additionally, our current inability to maintain aerosol under such high RH conditions limited the laboratory calibration and verification of this method using substances with known hygroscopic parameters (Fierz-Schmidhauser, et al., 2010; Zieger et al, 2013; Han et al., 2022), even though this method is theoretically feasible." **(Line 236-240)**

**#10.** Citations for works mentioned above:

Pope, F. D., Dennis-Smither, B. J., Griffiths, P. T., Clegg, S. L., & Cox, R. A. (2010). Studies of single aerosol particles containing malonic acid, glutaric acid, and their mixtures with sodium chloride. I. Hygroscopic growth. *The Journal of Physical Chemistry A*, *114*(16), 5335-5341.

AzadiAghdam, M., Braun, R. A., Edwards, E. L., Bañaga, P. A., Cruz, M. T., Betito, G., ... & Sorooshian, A. (2019). On the nature of sea salt aerosol at a coastal megacity: Insights from Manila, Philippines in Southeast Asia. *Atmospheric Environment*, *216*, 116922.

**R: Thank you for these references. Both of them are cited in the revised version.**

---

## Author Comment (AC2)

**Reply to Referee #1**

This paper reports on the use of what the authors claim is a simplified experimental system based on two nephelometers to determine the hygroscopicity parameter (Kappa) of ambient PM1 organic aerosol (Kappa_OA) for RH in the range of 85-95%. The system is essential two relatively low price nephelometers that are used to measure aerosol mass concentration, both measuring ambient air one at close to ambient conditions the other is dried. The results are contrasted for different aerosol sources and include comparison to the AMS-measured O/C ratio. The nephelometers also report mass concentration and so difference in the wet and dry nephs reported particle mass concentrations are interpreted as equal to the liquid water concentration due to the differences in the RH of the two nephs. There are some limitations noted by the authors, such as differences in particle size ranges when comparing masses from the dry neph to the AMS, that the AMS is not a comprehensive measurement of even PM1 mass, and uncertainty in the calibration of the nephs for converting scattering to mass. Furthermore, the sampling is done within an (I assume) airconditioned trailer which will result in biases when trying to determine actual ambient particle water concentrations, although that is not the goal of this study. For someone who has not read the first Zang et al paper on the pDRs, what these instruments actually are is not clear. Maybe a photo in the Supp, or a small description of what they are typically used for and stating the cost ($10k) early in the manuscript, not just in the Conclusions, would help to explain why this is claimed to be a simple method early in reading the paper.

One major issue lacking in this paper is a discussion comparing the specific method used here to the f(RH) method to infer particle water. Both use a wet and dry neph. The f(RH) method has a substantial history, yet is never noted in this work (see description in Guo et al and a list of references therein; www.atmos-chem-phys.net/15/5211/2015/)

Overall, the paper is of interest and suitable for publication in ACP but there are unclear sections in this paper that need to be addressed.

**R: We thank the reviewer for the detailed, helpful, and overall supportive comments. We have revised the manuscript to account for each comment. Responses to the individual comments are provided below. Below is our point-by-point response to each comment. Author responses are in Bold black. Modifications to the manuscript are in our normal font. Line numbers in the response correspond to those in the revised manuscript text file (tracked version).**

**For this major concern, we add more discussion to compare this specific methos to the previous studies, as "**A combination of dry and wet nephelometers has been used to estimate (1) aerosol liquid water content (ALW) (Guo et al., 2015; Kuang et al., 2018) and hygroscopicity (Kuang et al., 2017), replying on the measured aerosol light scattering enhancement factor ($f_{RH}$) (Fierz-Schmidhauser, et al., 2010; Titos, et al., 2016). When combined with aerosol chemical composition data, this approach also allows for the determination of $\kappa_{OA}$ (Kuang et al., 2020;

Kuang et al., 2021). These advancements have significantly promoted the application of nephelometers in aerosol hygroscopicity studies, and they also open up possibilities for using currently very popular, inexpensive optical scatter particle monitors for same purpose (e.g., Thermo pDR-1500, priced around $5,000; even more affordable options like Purple Air, costing a few hundred dollars, and Plantower PMS series, available for tens of dollars). These inexpensive devices, based on single-wavelength nephelometric technology, could potentially be used to infer aerosol hygroscopicity and associated ALW. However, unlike the commonly dry/wet nephelometers that measure particle scattering coefficients to calculate $f_{RH}$, these inexpensive particle monitors directly report particle mass concentration as a bulk measurement, essentially functioning as "black boxes". Unfortunately, there are very few studies that explore the potential of these optical particle monitors for such applications. " **(Line 50-61)**

**Thank you again for this comment, which has helped strengthen the manuscript.**

**Specific comments**

**1. In section 2.2 System setup, lines 93 to 100 where particle losses in sample lines are discussed it would be useful to add the flow Reynolds numbers. For line 99, what particle sizes does this less than 1% loss apply to?**

**R: Thank you for this comment. We added these related information, as "**The air flow was expected to be turbulent based on the calculated Reynolds Number (RN=30234, as determined from https://www.omnicalculator.com/physics/reynolds-number), and the estimated particle loss of the ambient aerosol, with a size between 100 nm to 1000 nm, from the van inlet to each instrument was less than 1% (https://www.mpic.de/4230607/particle-loss-calculator-plc).**" (Line 116-119)**

**2. Line 91, is RH of 45% sufficient to assume that particles do not contain water, which is, I believe, the assumption here in this calculation?**

**R: Thank you for this comment. We added more information, as "**During the deployment, the RH in pDR$_{dry}$ ranged between 30% and 45%. We used 45% as the upper RH threshold for self-calibration, based on the following considerations: (1) ISORROPIA II model calculations indicate that aerosol liquid water associated with inorganics (ALW$_{IOA}$) is zero for all data below 45% RH, and (2) submicron internally mixed inorganic-organic particles do not exhibit hygroscopic growth until they reach their deliquescence point, which occurs at approximately 77% RH (Pope et al., 2010; Jing et al., 2016; Bouzidi et al., 2020).**" (Line 105-109)**

**3. Line 118, what about the fact that the AMS only measures non-refractory species, so it is not**

a comprehensive measurement of particle mass concentration, not even considering the size of particles sampled. Ie, this should also be noted in this part of the paper, since it is also discussed later on, along with the PM$_1$ vs PM$_{2.5}$ issue.

**R: Thank you for this comment. We added more information, as "**Aside from the uncertainty due to size differences, the AMS only measures non-refractory aerosols and has limited sensitivity to refractory aerosols (e.g., sea salt), which introduces additional uncertainty and will be discussed further in Section 2.3." (Line 160-162)

**#4.** Line 140, why is the chemical composition data not used to estimate density of OA instead of assuming a constant value of 1.4 g/cm$^3$.

**R: Thank you for this comment. As not accurate species information of OAs, we used the 1.4 g cm$^{-3}$ for ρ$_{OA}$ following the commonly used value. We added more discussion about this, as:** "In this study, we used 1.4 g cm$^{-3}$ for ρ$_{OA}$ following the commonly used value (Hallquist et al., 2009; Shakya and Griffin, 2010; Nguyen et al., 2016; Riva et al., 2017; Jiang et al., 2019). However, the ρ$_{OA}$ can vary significantly depending on the sources and formation pathways of organic aerosols, with a range between 1.2 and 1.6 g cm$^{-3}$ based on a recent chamber study (El Mais et al., 2023), introducing some uncertainty into our results." **(Line 191-195). Thanks for your understanding.**

**#5.** Line 145, note that if these data are used to estimate ambient air LWC in this study there are issues with the ambient measurements (wet) being made indoors. This is why many past studies on using HTDMA or f(RH) run the ambient (wet) instrument outdoors.

**R: Thank you for this comment. Agree with this, and we add more information into the text, as "**Meanwhile, it also emphasized the possibility of using this system for using direct ambient measurements, very similar to the innovative outdoor dry/wet nephelometer system described by Qiao et al. (2024), without drying aerosols first before analysis as the HDMA (Tang et al., 2019) and without worrying about altering their actual phase state in ambient air (Qiao et al., 2024)." **(Line 200-204)**

**#6.** Line 150, what is the basis for assuming a constant fine/coarse mode mass ratio? Doesn't the fine and coarse mode chemical composition vary? Not sure how one assesses the impact of this assumption. The reasoning in lines 149 to 152 (" By simply assuming a constant …") is not clear. My interpretation is that the authors assume that the chemical composition of the coarse and fine modes is the same and invariant throughout the study and so the ratio of particle water in the fine and coarse modes will equal the ratio of fine and coarse mode dry mass concentration. This assumes no nonlinearities, such as the Kelvin effect.

**R: Thank you for this comment. We reword them as: "**The basics assumption here is the chemical composition of the coarse and fine modes is similar to each other throughout the study (Sun et al., 2020), and the ratio of particle water in the fine and coarse modes will equal the ratio of fine and coarse mode dry mass concentration. So that, the estimated $M_{ALW}$ here based on the calibrated aerosol mass concentration from the pDRs using AMS as reference can represent the liquid water in non-refractory $PM_1$. However, significant uncertainty will be introduced in the estimation of $\kappa_{OA}$, particularly due to the presence of sea salt and other high-$\kappa$ refractory components in coarse aerosols (AzadiAghdam et al. (2019)), which can greatly increase their hygroscopicity. Due to the limited information on the chemical composition (including refractory components) of fine and coarse aerosols, we can only provide a rough estimate of this uncertainty as a bulk, as shown below." **(Line 208-216)**

**#7.** Line 163, the standard deviation is given as 0.08, but this is somewhat meaningless without knowing the typical (mean) Kappa_OA. Maybe the range in the standard deviation divided by the mean could be given for all the bins to get an idea of the relative error estimated by this method. (do some calculation, to add more)

**R: Thank you for this comment. Considering the uncertainty on a quantity is generally quantified in terms of the standard deviation, we used the maximum standard deviation of all bins to represents the uncertainty for this method. For clarify, we add information of mean value, as "**The maximum standard deviation of $\kappa_{OA}$ across all bins of the identified three groups was determined to be 0.08 with the mean value of $\kappa_{OA}$ for this bin as 0.18, which was expected as the upper limit of the uncertainty for $\kappa_{OA}$**." (Line 230-232). Thank you so much for your understanding.**

**#8.** In Fig 2b define what the given ratios are (slope?). The associated text is not clear (lines 172-174, ie what is the 2.5 referring to, and [24].

**R: Thank you for this comment. We add the related information to Figure 2 caption, as** "The dashed lines represent the ratio lines of $PDR_{wetc}$ to $PDR_{dryc}$ at 1:1, 1.3:1, 2:1, and 4:1". **Meanwhile, we have revised the previous statement, as** "As shown in Fig. 3b, the mass growth factor ($=M_{pDRwetc}/M_{pDRdryc}$) was mainly between 2 to 4 under RH range of 90% to 100%, with an averaged value of 2.5, which was which was generally higher than the value under the RH range of 80% to 90% with an averaged value of 1.3." **(Line 262-265)**

**#9.** Fig 3, the x-axis has no label. This is somewhat stated in the fig caption but seems poor form. What is the year? Are the data shown in Fig 3 added (stacked) or each (ALW_OA and ALW_IOA) go to zero on the y axis? (correct it)

**R: Thank you for this comment. We add the x-axis label. It is a stacked column plot, and**

**we added it in the caption.**

[Figure]

P1: 89 points on June 26, 27, 28, 29 and July 02, 03, 04
P2: 14 points on July 05, 06
P3: 33 points on July 13, 14, 15
P4: 17 points on July 19, 20
P5: 13 points on July 21
P6: 13 points on July 21, 22

P7: 41 points on July 26, 27, 28, 29
P8: 30 points on Aug. 04, 05, 07
P9: 28 points on Aug. 10, 11, 12, 13, 14
P10: 38 points on Aug. 14, 15, 17, 18
P11: 32 points on Aug. 25, 26, 29
P12: 21 points on Sep. 04, 05, 06,07

**Figure 4. The time series of stacked column of the ALW$_{OA}$ and ALW$_{IOA}$ with a time resolution of 1-hr. (P1-P12 denote the different sub-periods mentioned in the following text with the data points and time periods of each subperiod indicated.)**

**#10.** Line 195, is derived Kappa_OA from equation 3, if so state it.

**R: Thank you for this comment. We added it, as "**The box and whiskers distribution of the derived $\kappa_{OA}$ based on Eq. (4) for each sub-period is shown in Fig. 5a**" (Line 290)**

**#11.** Typo in line 200 ,,

**R: Thank you for this comment. We corrected it.**

**#12.** Line 196 and Fig 4b, define mass concentration, ie is it dry PM1? (Not sure what total mass concentration means).

**R: Thank you for this comment. We added the information, as "**alongside the HR-ToF-AMS measured PM$_1$ mass concentration in Fig. 5b". **(Line 291)**

**#13.** Line 214 to 216. Doesn't burning conditions, smoldering/flaming affect Kappa_OA, or is

this washed out the in highly averaged nature of smoke transported over long distances?

R: Thank you for this comment. We added the information, as "it is reasonable to infer the wildfire $\kappa_{OA}$ could be strongly affected by the burning time of the original forests, the related burning conditions (i.e., smoldering vs. flaming, etc.), the transport time from west to east, etc. (Garofalo et al., 2019), resulting in significant variation between different cases, warranting further investigation." (Line 310-312)

**14. Would it be useful to plot Kappa_OA to Mass_ALWOA? They are related by equation 3.**

R: Thank you for this comment. As they are directly related to each other based on new Eq. (4), we did not to plot it again. Thank you so much for your understanding .

**15. Line 237 starting with " It also shows…. What is being referred to, Fig 5b? (change to: It shows to Fig 5b shows…?**

R: Thank you for this comment. We corrected it, as "Meanwhile, the near-constant trends of $\kappa_{OA}$ are showed for each period affected by the wildfire plumes (Fig. 6b)" (Line 334-335)

**16. First line of Conclusions, why not call them inexpensive single wavelength nephelometers instead of optical scattering systems, the latter could include a single particle optical particle counter, which these are not (I assume). (not hard)**

R: Thank you for this comment. We corrected it from the comment.

**17. Line 263, not only is the slope different but the magnitude is significantly different between urban and rural (the curves are nowhere near overlapping). Doesn't this have implications for using O/C to estimate Kappa_OA.**

R: Thank you for this comment. We added this information as "Meanwhile, the magnitude of $\kappa_{OA}$ of rural aerosol is much higher than the value of urban aerosol." (Line 361)

and,

"These large different $\kappa_{OA}$ vs. O:C relationships, including both slopes and magnitudes, for each group imply the necessity of estimation of $\kappa_{OA}$ through direct measurements, rather than through a simple dependent relationship based on one kind of aerosol other properties (i.e., O:C ratio)." (Line 364-366)

**#18.** Line 279, typo, varication?

**R: Thank you for this comment. We corrected it, and moved it to new section 2.3, as "**It is important to note that the derived $\kappa_{OA}$ values in this study were not continuous, as we could only obtain them under high relative humidity (RH) conditions (85% to 90%). Additionally, our current inability to maintain aerosol under such high RH conditions limited the laboratory calibration and verification of this method using substances with known hygroscopic parameters (Fierz-Schmidhauser, et al., 2010; Zieger et al, 2013; Han et al., 2022), even though this method is theoretically feasible." (Line 236-240)

**#19.** A final comment: It is curious to me why one does not compare water soluble organic carbon to Kappa_OA. (no hard)

**R: Thank you for this comment. During the field measurements, we did not have an instrument (i.e., PILS) to report the water soluble organic carbon. Thanks for your understanding.**

**#20.** Lines 277 and on where it is noted that there the measurements were not continuous…. This is not clear. The schematic shows that the wet measurement was straight ambient. It then seems that the gaps in the data are due to only periods of high ambient RH were analyzed in this study. So the authors are suggesting that adding a humidification system to the ambient leg to maintain an RH in a specific range, such as 85-95% would allow continuous measurements – is this the point? (make it more clear)

**R: Thank you for this comment, and sorry for the confusion. You are totally right, and we rewrite it as, "**It is important to note that the derived $\kappa_{OA}$ values in this study were not continuous, as we could only obtain them under high relative humidity (RH) conditions (85% to 90%). Additionally, our current inability to maintain aerosol under such high RH conditions limited the laboratory calibration and verification of this method using substances with known hygroscopic parameters (Fierz-Schmidhauser, et al., 2010; Zieger et al, 2013; Han et al., 2022), even though this method is theoretically feasible. To resolve this issue, one possible update of this system could be adding a humidifier system to the pDR to get wet aerosol with RH between 85% to 95%, and the possible set-up for humidifier system could include a Perma Pure MH-series humidifier, water pumps and tanks (red dash box in Figure 2). This will make this system more be similar to the widely used humidified nephelometer system (Guo, et al., 2015; Burgos et al., 2019, Fierz-Schmidhauser et al., 2010; Kuang et al., 2017,2018,2020, 2021)." (Line 236-244).

---

## Author Comment (AC3)

**Reply to Dr. Ye Kuang**

**Dear Dr. Kuang,**

**Thank you so much for your comments. Based on your suggestions, we add more discussion into the text with all references are cited, as**

"A combination of dry and wet nephelometers has been used to estimate (1) aerosol liquid water content (ALW) (Guo et al., 2015; Kuang et al., 2018) and hygroscopicity (Kuang et al., 2017), replying on the measured aerosol light scattering enhancement factor ($f_{RH}$) (Fierz-Schmidhauser, et al., 2010; Titos, et al., 2016). When combined with aerosol chemical composition data, this approach also allows for the determination of $\kappa_{OA}$ (Kuang et al., 2020; Kuang et al., 2021). These advancements have significantly promoted the application of nephelometers in aerosol hygroscopicity studies, and they also open up possibilities for using currently very popular, inexpensive optical scatter particle monitors for same purpose (e.g., Thermo pDR-1500, priced around $5,000; even more affordable options like Purple Air, costing a few hundred dollars, and Plantower PMS series, available for tens of dollars). These inexpensive devices, based on single-wavelength nephelometric technology, could potentially be used to infer aerosol hygroscopicity and associated ALW. However, unlike the commonly dry/wet nephelometers that measure particle scattering coefficients to calculate $f_{RH}$, these inexpensive particle monitors directly report particle mass concentration as a bulk measurement, essentially functioning as "black boxes". Unfortunately, there are very few studies that explore the potential of these optical particle monitors for such applications. " **(Line 50-61 in the revised version)**

**And,**

"Meanwhile, it also emphasized the possibility of using this system for using direct ambient measurements, very similar to the innovative outdoor dry/wet nephelometer system described by Qiao et al. (2024), without drying aerosols first before analysis as the HDMA (Tang et al., 2019) and without worrying about altering their actual phase state in ambient air (Qiao et al., 2024)."
**(Line 200-203 in the revised version)**

---

## Author Comment (AC4)

**Reply to Dr. Paul Zieger**

**Dear Dr. Zieger,**

**Thank you so much for your comments, and we totally agree that lab calibration and verification of this method using the substances with known hygroscopic parameters is critical important. We proposed a potential design for this type of test, but it could not be realized in our lab at this time due to limited resources. To made up this to some extent, we add more detailed discussion in the text with all reference being cited and including the proposed instruments set-up, as**

"It is important to note that the derived $\kappa_{OA}$ values in this study were not continuous, as we could only obtain them under high relative humidity (RH) conditions (85% to 90%). Additionally, our current inability to maintain aerosol under such high RH conditions limited the laboratory calibration and verification of this method using substances with known hygroscopic parameters (Fierz-Schmidhauser, et al., 2010; Zieger et al, 2013; Han et al., 2022), even though this method is theoretically feasible. To resolve this issue, one possible update of this system could be adding a humidifier system to the pDR to get wet aerosol with RH between 85% to 95%, and the possible set-up for humidifier system could include a Perma Pure MH-series humidifier, water pumps and tanks (red dash box in Figure 2). This will make this system more be similar to the widely used humidified nephelometer system (Guo, et al., 2015; Burgos et al., 2019, Fierz-Schmidhauser et al., 2010; Kuang et al., 2017,2018,2020, 2021).

[Figure]

**Figure 2. proposed instrument setup for lab calibration**

The proposed instrument setup for lab calibration will include an atomizer to produce aerosol, which will be dried through the dryer. Then, the Differential Mobility Analyzer (DMA) will pick up different size of aerosols, with one of branch aerosol flow to the humidifier system to get wet and then to pDRwet, and another three branches of aerosol flows to pDR$_{dry}$, AMS, and condensation particle counter (CPC). The substances/aerosol for testing will include the organic aerosols with known hygroscopic parameter (Han et al., 2022), the inorganic aerosols (i.e., $(NH_4)_2SO_4$, Fierz-Schmidhauser et al., 2010), as well as their mixing solutions of organic and inorganic. Due to limited resources, this proposed instrument setup is not feasible at this moment, and the lab calibration is not included in this study. However, we hope this will inspire other research groups with this set-up to conduct these lab tests to better quantify the uncertainty of this method for pDRs. Given that the pDR is a type of single-wavelength nephelometric monitor, it's logical to consider that other brands commonly used low-cost nephelometric monitors (e.g., Purple Air, Plantower PMS series) might offer similar capabilities, and related lab test would be also highly recommended.**" (Line 236-257 for the revised version). Thank you so much for your understanding.**

---

## Author Response (AR2)

**Reply to Referee #1**

**We sincerely thank the reviewer for their support in recommending the acceptance of our manuscript for publication. We corrected the typo, and change "replying on" to "relying on".**